# Effect of Catechin on Yolk Immunoglobulin Structure and Properties: A Polyphenol–Protein Interaction Approach

**DOI:** 10.3390/foods12030462

**Published:** 2023-01-19

**Authors:** Lili Liu, Xiaodan Zhang, Mapinyi Zhang, Mengjun Zhang, Weiwei Cheng, Baocheng Xu

**Affiliations:** College of Food and Bioengineering, National Experimental Teaching Demonstration Center for Food Processing and Security, International Joint Laboratory of Food Processing and Quality Safety Control of Henan Province, Henan Engineering Technology Research Institute of Food Microbiology, Henan University of Science and Technology, Luoyang 471023, China

**Keywords:** catechin, yolk immunoglobulin, structure, properties, interaction

## Abstract

The preparation of the interaction between polyphenols and protein is of great significance for increasing added value and promoting the application of egg yolk immunoglobulin (IgY). This study systematically investigated the effect of catechin on yolk immunoglobulin structural characteristics and functional properties. The binding conditions, force types, molecular conformation, and residual microenvironment of the interaction between catechin and IgY were analyzed by molecular docking technology, UV-vis absorption and fluorescence spectroscopy studies. The results showed that the main binding forces in the complex were hydrogen bonding and van der Waals forces. After the interaction, fluorescence quenching occurred and the maximum emission wavelength was redshifted. The results showed that the microenvironment around IgY increased polarity, increased hydrophilicity and decreased hydrophobicity, and the structure of the peptide chain changed. The bacteriostatic thermal stability of the compound against *Escherichia coli* and *Staphylococcus aureus* was lower than that of catechin IgY. The bacteriostatic acid and base stability were higher than that of catechin and IgY. The antioxidant activity was catechin, complex, and IgY, in descending order. The antioxidant activity of catechin and complex was significantly higher than that of IgY. At the same concentration, the apparent viscosity of the three samples was complex, IgY and catechin, in descending order. G’ was greater than G” indicating that elastic properties dominate in G”. The G’ and G” values of the complex were higher than those of the other groups. Rheological results indicated that the complex may have high physical stability. This study provides theoretical support for broadening the application field of IgY and suggest its properties change in the machining process. It also provides new ideas for the development of functional foods from poultry eggs.

## 1. Introduction

IgYs are the main constituent of γ-livetin, and among the most important and most abundant egg yolk proteins [1]. IgY can be isolated from “immune” egg yolk in large quantities in order to avoid animal bleeding, which causes stress on animals, and has been applied in various fields of biotechnology and biomedicine [2,3,4]. A large number of studies have found that IgY has immunological properties, such as not binding to human complement and rheumatoid factors, allowing it to be applied in many areas such as immunological diagnosis and medicine [5]. However, IgY has not been widely used as an ideal product. This is partly due to the technical difficulties involved in isolating IgY from egg yolk and its susceptibility to temperature, acid and base. IgY has the lowest stability of all immunoglobulins, thus its application field is limited [6]. China is rich in egg resources, so it is of great economic and practical significance to make full use of chicken yolk resources to extract immune active components and protect their stability [7].

The physical and chemical properties of IgY directly affect its preparation and application in various fields, such as heat resistance, acid and alkali resistance [8]. Results show that IgY has a relatively stable activity over a certain range of heat treatment and acid–base treatment. However, when the temperature is too high, or the pH is too low, or too high, its activity will be significantly reduced and irreversible inactivation will occur [9]. Existing studies have confirmed that coating IgY with liposome or chitosan–sodium alginate can significantly improve this problem and improve the acid–base stability of IgY [10]. However, there are also practical problems as the production process becomes complicated and the production cost increases when the IgY is coated, which is not conducive to large-scale application. In recent years, the interaction of polyphenols with proteins to prepare complexes to improve their functional properties has become a hot topic.

A large number of studies have shown that polyphenols can interact with proteins to form reversible or irreversible complexes, thus affecting the functional properties and bioavailability of proteins [11]. Jia et al. have analyzed the effect of covalent interaction between EGCG and whey protein isolation on its functional properties, and the results showed that its foaming and emulsifying properties were improved [12]. Zhao et al. have shown that tea polyphenols (TP) are important in the post-mortem deterioration of fish muscle and can enhance food quality [13]. Chaijan et al. have investigated the effects of whey protein isolate (WPI)–polyphenol coating on the microbial counts, physico-chemical properties, and sensory qualities of Asian sea bass (*Lates calcarifer*) steak, during chilled storage and found that coating with WPI–phenolic extracts was an effective method for the preservation of chilled Asian sea bass steak [14]. Many studies have shown that the interaction between polyphenols and protein can optimize the characteristics of most proteins. However, research on modification of IgY to improve its properties has mainly focused on the effects of glycosylation and mPEG modification on its stability [15]. There are few reports on the interaction between polyphenols and IgY to prepare complexes and explore the changes of its structural and functional characteristics. Zhang et al. have studied the interaction mechanisms between mung bean protein and polyphenols in a water solution system of heat treatment and speculated on the binding form of the two [16]. Polyphenol addition improved mung bean globulin secondary structure, and the thermal stability of the complex was greater than that of mung bean globulin.

Therefore, in this study, molecular docking technology and three-dimensional fluorescence spectroscopy were used to analyze the binding interaction of catechin and IgY, in terms of the type of interaction force, and the changes in molecular conformation and residue microenvironment of the system after binding. The complex was prepared by the interaction of catechin and IgY. The thermal stability, acid–base stability, antioxidant activity and rheological properties of IgY-catechin complex were studied with IgY and catechin as controls. We provide a theoretical basis for the application of IgY in functional foods and its property changes during processing, and provide a functional food base material for the health product market. This work has important research significance and application value.

## 2. Materials and Methods

### 2.1. Materials

Fresh eggs were purchased from Dazhang Supermarket (Luoyang, China). DPPH (1, 1-diphenyl-2-picrylhydrazine radical) was from Blue Season organisms Co., Ltd. (Shanghai, China), FeSO_4_ and salicylic acid were purchased from Deen Chemical Reagent Co., Ltd. (Tianjin, China), 30% H_2_O_2_ was purchased from Haohua Chemical Reagent Co., Ltd. (Luoyang, China), *E. coli* and *S. aureus* were obtained from the microbiology lab of the Food and Bioengineering college, Henan University of Science and Technology. All chemicals used were of analytical grade. All solutions and suspensions were prepared with deionized water.

### 2.2. Preparation of IgY and IgY-Catechin Complex

Following the method described in detail by Nanase et al. [17], IgY solution was prepared by ammonium sulphate precipitation. Subsequently, the prepared IgY solution was dialyzed at 4 °C (molecular weight of 8000–14,000 Da) for 48 h, and then vacuum freeze-dried IgY powder was prepared. IgY and catechin were mixed at a mass concentration of 1:1, stirred in a magnetic agitator for 6 h at room temperature, and the complex particles were prepared by vacuum freeze drying.

### 2.3. Molecular Docking Study

The structure of IgY (PDB ID: 2w59) was obtained from the RCSB protein databank website (rcsb.org). Catechin was obtained from the PubChem databank (https://pubchem.ncbi.nlm.nih.gov, accessed on 28 October 2022). AutoDockTools-1.5.6 software was employed to establish the interaction molecular model. The Discovery Studio Visualizer 2018 software was employed to acquire better visualization of the docked conformations between IgY and catechin. The best-scoring pose was analyzed and drew the simulation results using PyMoL software (DeLano Scientifific LLC, South San Francisco, CA, USA).

### 2.4. UV-Vis Absorption Study of the Interaction between Catechin and IgY

IgY was prepared into a 0.5 mg/mL solution, and 4 mL of each solution was transferred to centrifuge tubes numbered 1–5, where 50 μL catechin solutions with different mass concentrations were added, to bring the final concentrations to 0, 6.25, 12.5, 18.75 and 25 μg/mL. After complete mixing, samples were allowed to stand for 20 min. The UV-vis absorption spectra of the interaction of catechin and IgY was investigated using a UV-2600 spectrophotometer (Hitachi Co., Ltd., Hitachi, Japan) at intervals of 5 nm in the range of 250~350 nm. The optical path was 1 cm and the response time was 0.2 s.

### 2.5. Fluorescence Spectroscopy of the Interaction between Catechin and IgY

The sample was prepared as per method 2.4 in a constant water bath for 5 min at 18 °C, 25 °C and 37 °C, respectively. The fluorescence spectra of the interaction between catechin and IgY was measured by using the Cary eclipse fluorescence spectrometer (America Aglient Cary elipse Co., Ltd., Palo Alto, CA, USA) at excitation wavelength (280 nm), excitation and emission slit (10 nm) and scanning range (300~500 nm).

### 2.6. Three-Dimensional Fluorescence Spectrometric Determination of the Interaction between Catechin and IgY

In this experiment, IgY was prepared as a 0.5 mg/mL solution, 4 mL of IgY was transferred to centrifuge tubes numbered 1 to 5, and 50 μL catechin solution of different mass concentrations were added, so that the final concentrations were 0, 6.25, 12.5, 18.75, 25 μg/mL, respectively. Solutions were mixed thoroughly and allowed to stand for 5 min. At 200~450 nm excitation (10 nm interval), the fluorescence emission spectra over the range of 200~500 nm was obtained from the IgY system before and after catechin was added.

### 2.7. Bacteriostatic Thermal Stability Analysis

The bacteriostatic thermal stability of the solution was determined according to the method of Li et al. [18], with slight modification. The samples were prepared as a 0.5 mg/mL solution and treated at different temperatures (40, 50, 60, 70, 80 and 90 °C) for 20 min. The Luria–Bertani (LB) liquid medium (4 mL) was placed in flasks. To the sterilized test tube, 100 μL bacterial solution and 1 mL of sample solution treated under different conditions were added. A total of 1 mL of sterilized distilled water and 100 μL bacterial solution were added as a positive control and only the corresponding samples were added as the negative control. The flask was incubated at 37 °C under shaking conditions for 12 h and the optical density (OD) was measured at 600 nm. The change in antibacterial growth rate was measured and the stability under different temperature conditions was explored.
(1)$$\mathrm{Bacteriostatic}\;\mathrm{rate}=\frac{\mathrm{OD}\;\mathrm{value}\;\mathrm{of}\;\mathrm{positive}\;\mathrm{control}-\mathrm{OD}\;\mathrm{value}\;\mathrm{of}\;\mathrm{test}\;\mathrm{group}}{\mathrm{OD}\;\mathrm{value}\;\mathrm{of}\;\mathrm{positive}\;\mathrm{control}-\mathrm{OD}\;\mathrm{value}\;\mathrm{of}\;\mathrm{negative}\;\mathrm{control}}$$

### 2.8. Stability Analysis of Bacteriostatic Acid and Base

The pH of the solutions were adjusted using an appropriate buffer to 5, 6, 7, 8, 9 and 10, respectively. The change of measured bacteriostatic rate according to the above method was used to explore the antibacterial stability under different pH conditions.

### 2.9. Analysis of Antioxidant Activity

#### 2.9.1. DPPH Free Radical Scavenging Rate

The method of Marina et al. [19] was adopted and slightly modified. Sample solutions of 0.1, 0.4, 0.6, 1.2, 1.6 mg/mL were prepared. A sample of 1.5 mL was mixed with 0.2 mmol/L DPPH-ethanol. The reaction time was 30 min under dark conditions at 25 °C. The absorbance was measured at 517 nm.
(2)$$\mathrm{DPPH}\;\mathrm{clearance}\;(100\%)=(1-\frac{A_{1}{-A}_{2}}{A_{0}})\times 100$$

A_0_ was the absorbance value of the mixture of 1.5 mL water and 1.5 mL DPPH-ethanol. A_1_ was the absorbance value of the mixture of 1.5 mL sample and 1.5 mL DPPH- ethanol. A_2_ was the absorbance value of the mixture of 1.5 mL sample and 1.5 mL ethanol.

#### 2.9.2. OH Free Radical Scavenging Rate

The method of Chen et al. [20] was slightly modified. The sample solutions with different concentrations were treated as follows: 1 mL of the sample was mixed with 0.2 mmol/L FeSO_4_ solution and 0.2 mmol/L H_2_O_2_. After reaction for 40 min at 25 °C using a water bath, the ultraviolet absorbance of the mixed solution was measured at 510 nm.
(3)$$\mathrm{OH}\;\mathrm{clearance}\;(100\%)=(1-\frac{A_{1}{-A}_{2}}{A_{0}})\times 100$$

A_0_ was the absorption value of the mixture of 1 mL water, 1 mL FeSO_4_, 1 mL salicylic acid and 1 mL H_2_O_2_. A_1_ was the absorption value of the mixture of 1 mL sample, 1 mL FeSO_4_, 1 mL salicylic acid and 1 mL H_2_O_2_. A_2_ was the absorption value of the mixture of 1 mL sample, 1 mL FeSO_4_, 1 mL ethanol and 1 mL H_2_O_2_.

### 2.10. Differential Scanning Calorimetry Analysis

Determination was performed according to Liu et al. [21]. The thermal denaturation temperature of egg white was measured by a differential scanning calorimeter. The temperature range was 25–140 °C, the heating rate was 10 °C/min, the sample dosage was 2–3 mg each time, and the nitrogen flow rate was 60 mL/min.

### 2.11. Rheological Analysis

IgY, catechin and IgY-catechin complex were prepared as a solution of 10 mg/mL, respectively. A 60 mm plate–plate was chosen. The apparent viscosity, elastic modulus G’ and viscous modulus G” of the samples were measured by a DHR-2 Rheometer (Waters Co., Ltd., Milford, MA, USA) at 25 °C in the angular frequency range of 0.1–100 rad/s.

### 2.12. Statistical Analysis

All measurements were performed in triplicate. The statistical significance of the data were analyzed using SPSS 22.0 (SPSS, Inc., Chicago, IL, USA). The analyses of variance were performed using ANOVA. The significance level of data was set at *p* < 0.05. The correlation analysis was carried out using Origin 9.0.

## 3. Results and Discussion

### 3.1. Molecular Docking Analysis

The molecular docking technology was used to simulate the interaction and more intuitively reflect the binding state of catechin and IgY. IgY was set as the receptor and catechin as the ligand. Molecular docking was used to simulate the interaction between ligand and receptor at the molecular level. The interaction between polyphenols and proteins is mainly non-covalent under neutral or acidic conditions due to polyphenols not easily forming quinone or semi-quinone radicals in these conditions. The results are shown in Figure 1. The binding site of catechin and IgY was the groove of IgY as the active center. Catechin formed hydrogen bonds with residues ALA-462 and GLN-563 in IgY. Yu et al. [22], studied the interaction mechanism of TPs-BSA by molecular docking and the results indicated EGCG was mainly surrounded by the residues GLN-32, GLN-33, THR-83, TYR-30, HIS-105, and ASP-107 via seven hydrogen bonds when binding with TYR-84 and hydrogen bonding was the main driving force under these experimental conditions.

### 3.2. UV-Vis Absorption Spectra Study

The binding of small molecule materials and protein causes the protein structure to change. The UV-vis spectrum can reflect how it interacts through the changes in the peak intensity and peak position. The changes in the peak intensity and peak position reflect the strength of the interaction and indicate the microenvironment of the hydrophobic amino acid residues of the protein macromolecules have altered, respectively. The tryptophan and tyrosine residues in the protein, have absorption peaks that occur near 280 nm in the UV-vis spectrum. The IgY concentration was set to 0.5 mg/mL, and the UV absorption spectrum of the two combinations were observed after adding different concentrations of catechin, and the spectral changes are shown in Figure 2. The greater the catechin concentration, the higher the peak UV absorption. The maximum absorption peak wavelength was blue shifted from 279 nm to 276 nm, indicating that the conformation of IgY had changed. The reason may be that the addition of catechin could change the peptide chain in IgY and make it more stretched, helping the protein to release aromatic rings in the chromatin group residues inside the molecule. Ye et al. [23] used UV-vis spectroscopy to provide information about the structure of rutin-SPI complexes and the results suggested that there was an interaction between rutin and SPI, which presumably involved the tyrosine and tryptophan groups and was attributed to polyphenol–protein interactions.

### 3.3. Fluorescence Spectroscopy

The chromogroups in protein molecules can produce characteristic peaks at specific wavelengths, amino acid residues in the chromogenic group that exhibit weak emission fluorescence are ignored, therefore, the fluorescence spectrum changes of tryptophan (348 nm) are mainly studied. The fluorescence spectra of catechin with different mass concentrations binding to IgY at 18 °C at an excitation wavelength of 280 nm was studied, and the result is shown in Figure 3. As a small molecule, catechin binds to IgY, which reduces the quantum yield of the emitted fluorescent substance in the protein molecule, reducing its fluorescence intensity and producing a fluorescence quenching phenomenon. Figure 3 shows the fluorescence spectra of catechin binding to IgY at different mass concentrations during treatments at different temperatures. According to Figure 3, the fluorescence quenching effect of IgY was enhanced and redshifted with increasing catechin concentration. It was further shown that the interaction between the two occurs, and that a complex with a lower fluorescence intensity was formed during the promotion of this interaction. The reason may be that the addition of catechins causes changes in the microenvironment around tryptophan in IgY, promoting hydrophobic interactions between the two and the formation of hydrogen bonds. It is important to note that heat treatments also had an effect on the fluorescence intensity of IgY. With IgY fluorescence intensity gradually decreasing in the temperature range from 18 to 37 °C, similar changes in the fluorescence quenching effect occur in the presence of catechins. Chen et al. [24] investigated the fluorescence spectroscopy of the interaction between soybean Bowman–Birk trypsin inhibitor (BBTI) and epigallocatechin, the results showed that the fluorescence intensity of the complex gradually decreased with increasing temperatures, which was consistent with the results of this paper.

### 3.4. Three-Dimensional Fluorescence Spectra

The three-dimensional fluorescence spectrum is an effective integration of the excitation spectrum and emission spectrum. It can provide more complete, rich, and valuable information for the determination and identification of samples with fluorescence effects. It can infer some information about whether the characteristic conformation of protein molecules and the microenvironment of luminescent groups have changed [25]. In order to further study the changes in conformation of the polypeptide skeleton and residual microenvironment after IgY and catechin binding, different concentrations of catechin were added to the IgY solution, and its three-dimensional fluorescence spectra was determined. The result is shown in Figure 4. There were mainly two kinds of peaks in the three-dimensional fluorescence spectrum. One was an ordered Rayleigh scattering peak, namely Peak A and Peak B (λex = λem) with similar shape to the ridge, and the other was typical fluorescence Peak 1 and Peak 2 (2λex = λem) with similar shape to a hump. The spectral characteristics of peak 1 were mainly Trp and Tyr residues, and peak 2 was mainly related to the fluorescence spectral behavior of the peptide skeleton structure, whose intensity was related to protein secondary structure [26]. As shown in Figure 4, with the increase of catechin concentration, the fluorescence intensity of Peak 1 and Peak 2 decreased significantly, and the maximum emission wavelength was red shifted. The decrease of fluorescence intensity indicated that catechin interacted with IgY and produced a fluorescence quenching phenomenon. The red shift phenomenon indicated that catechins affected the microenvironment around IgY. That is, polarity and hydrophilicity increased, hydrophobicity decreased, and the structure of the peptide chain changed. Cao et al. [27] investigated how CGA quenched intrinsic fluorescence intensities of enzymes by static quenching and binding with CGA which led to changes in 3D structures of enzymes.

### 3.5. Effects of IgY and Catechin on Antibacterial Thermal Stability

Using IgY and catechin as the controls, and the antibacterial rate as the index, the changes of antibacterial thermal stability of the catechin -IgY complex to *E. coli* and *S. aureus* at different temperatures were compared and analyzed. The results are shown in Figure 5. Under the same conditions, the antibacterial rate of the three samples against *E. coli* showed complex > catechin > IgY. With the temperature increasing, the antibacterial rate of all three samples against *E. coli* decreased. The antibacterial activity of IgY against *E. coli* decreased significantly above 70 °C (*p* < 0.05), the antibacterial activity of catechin against *E. coli* decreased significantly above 80 °C (*p* < 0.05) and the complex was significantly reduced at temperatures above 60 °C (*p* < 0.05). Compared with IgY and catechin, the antibacterial thermostability was reduced against *E. coli*. As Figure 5b shows, under the same conditions, the three samples showed catechin > complex > IgY. With the temperature increasing, the antibacterial rate of catechins and IgY against S. aureus was significantly reduced above 70 °C (*p* < 0.05), while the complex decreased significantly at temperatures above 60 °C (*p* < 0.05). Compared with IgY, the antibacterial thermal stability of the complex to *S. aureus* decreased. The reason for these phenomena may be the interaction between catechin and IgY, which changes the original secondary structure of the protein macromolecule, exposes more active groups and enhances antibacterial activity. In addition, the temperature affected the stability of the compound to a certain extent. When the temperature rose, the molecular force between them was weakened or destroyed, which decomposed the compound, resulting in a decrease of antibacterial stability. Hajarian et al. [28] revealed that the algae–chitosan mixture extract-assisted AgNPs induce superior effectiveness for all selected gram-positive and gram-negative bacterial strains in comparison to the corresponding commercial silver nanoparticles, initial algae–chitosan extract, chitosan, algae extract, and silver nitrate precursor, respectively.

### 3.6. Effects of IgY and Catechin on Stability of Bacteriostatic Acid and Base

The bacteriostatic acid–base stability of IgY, catechin and the IgY-catechin complex against *E. coli* and *S. aureus* were analyzed by different pH treatments. The result is shown in Figure 6. According to Figure 6a, under the same conditions, the IgY-catechin complex had the highest bacteriostatic rate against *E. coli* compared with other samples. Under different pH conditions, the change trend of the complex curve was relatively flat. Compared with IgY and catechin, the complex had a higher stability of bacteriostatic acid and base to *E. coli*. According to Figure 6b, under the same conditions, the antibacterial rates of the complex and catechin against *S. aureus* were significantly higher than IgY, but there was no significant difference between the two. The change trend of the complex and catechin against *S. aureus* was basically consistent with pH value. The bacteriostatic activity decreased significantly when pH was higher than 8. Compared with the IgY, complex and catechin it showed better bacteriostatic acid–base stability against *S. aureus*. The reason may be that IgY and catechin interact and combine to form the complex, which reduces the ionization degree of the hydroxyl group in catechin, and makes it easier to bind with cell membranes and lipoproteins of the bacterial body after interaction, synergistically enhancing the antibacterial activity and further improving the stability of antibacterial acid and base. Wu et al. [29] prepared curcumin-loaded nanoliposomes from bovine milk phospholipid and cholesterol, and found that cholesterol helps formation of a more hydrophobic, compact, and tighter bilayer membrane structure to improve the storage stability of nanoliposomes under alkaline, heat, and sunlight conditions.

### 3.7. Effect of the Interaction between IgY and Catechin on Antioxidation

The antioxidant effect of protein is related to its structural characteristics such as chain conformation and molecular weight. The scavenging capacity of IgY, catechin and the complex on the DPPH free radical and the hydroxyl free radical was determined, and the changes of antioxidant capacity after the interaction of catechin and IgY were compared and analyzed. The result is shown in Figure 7. As can be seen from Figure 7, with the increase of IgY concentration, its free radical scavenging ability shows an increasing trend. The hydroxyl radical is the most reactive species among reactive oxygen species. OH can react with all amino acids and consequently polyphenol is a good candidate as an OH scavenger. Compared with the DPPH radical, the scavenging ability of IgY against the hydroxyl radical increased slowly with the increase in concentration. The antioxidant capacity of IgY was low. After catechin was added, the two interacted to form a complex and the antioxidant capacity of the complex was significantly increased. Therefore, interactions between IgY and catechin resulted in increased antioxidant capacity of the complex, probably due to the change in the protein structure. However, compared with catechins under the same conditions, the antioxidant capacity of the complex was lower. The reason may be that some phenolic hydroxyl groups of catechin bind to IgY or are embedded in the protein, which reduces the probability of collision with free radicals and results in a lower antioxidant capacity than catechin. Yuan et al. [30] showed that the antioxidant capacity of the films, as determined by DPPH and ABTS radical analysis, and the anti-oxidation and anti-inflammatory properties were enhanced by incorporating tea polyphenols into them.

### 3.8. Effect of IgY and Catechin on Rheological Properties

Compared with IgY and catechin, the IgY-catechin complex showed higher apparent viscosity at a certain shear rate according to Figure 8. It indicated that the protein system after IgY and catechin interaction tends to be stable. The reason may be that IgY interacts with catechins to form a complex with a larger molecular weight, which increases flow resistance and viscosity. According to Stokes’ law, the greater the internal resistance encountered by the solution in the process of flow, the greater the obstacle to the free movement of the medium, thus showing a higher apparent viscosity. The higher the viscosity of solution particles, the slower the precipitation rate, the more stable the liquid. Therefore, based on the different rheological properties of the above samples, it was suggested that the composite may have high physical stability. In addition, the apparent viscosity of all samples decreased with the increase of angular frequency. All samples showed shear dilution, which accords with non-Newtonian fluid characteristics. Small oscillatory deformation rheological tests have been widely used to study viscoelastic and molecular interactions in protein gels. It can be seen from Figure 8 that G’ and G” of the three samples increased with the increase in frequency, and the viscoelasticity was frequency dependent. It indicated that the weak gel formed by the samples produced relaxation phenomenon. G’ was greater than G” and this indicated that elastic properties dominate. The G’ and G” values of the complex were higher than those of other groups. These results suggested that the interaction between catechins and IgY could promote the formation of uniform, dense, and high-strength protein gel network structures to a certain extent [31]. Compared with catechin and IgY, G’ of the complex increased, which may be due to the interaction between IgY and catechin, resulting in changes in protein structure. A new interaction force was formed between them, which further affected the covalent interaction between protein molecules and the gel network structure.

### 3.9. DSC

The results of DSC analysis of the interaction between IgY and catechin are shown in Figure 9. As the result of Figure 9 shows, the DSC thermal denaturation temperature curves of the two samples had obvious exothermic peaks, because the protein conformation had changed due to thermal denaturation, the maximum denaturation temperature of IgY, catechin and the complex were 73.79 °C, 115.47 °C and 60.94 °C, respectively. Compared with IgY and catechin, the maximum denaturation temperature (Tm) of the complex was the lowest. The thermal stability of proteins was related to their amino acid composition, and the thermal stability of proteins with a higher ratio of hydrophobic amino acids was higher than proteins with a lower ratio of hydrophilic amino acids generally. The reason may be that the combination of catechin and IgY affected the hydrophobic and electrostatic interactions of the protein molecules, the IgY globular structure was destroyed by catechin, which resulted in a decrease of thermal stability. The result is in line with a study on the interaction of squid mantle protein with a mixture of potato and corn starch explored by Luis et al. [32]. The Tm values in extruded samples were directly proportional to the squid content as the amine groups I and II, responsible for the protein–protein interaction, were reduced and the O-glucosidic bonds were increased.

## 4. Conclusions

In summary, this study investigated the complex of catechin and IgY that was successfully prepared by molecular docking. The decrease in fluorescence intensity indicated that catechin interacts with IgY and causes fluorescence quenching. The microenvironment around IgY changes, and the polarity is enhanced, hydrophilicity is increased, hydrophobicity is decreased, and the secondary structure changes. In addition, compared with the two monomers, the complex had better antibacterial stability and antioxidant properties, which could be used in food and medical applications. Rheology and DSC analysis further verified the effects of the interaction between them on the structure and properties. Therefore, the combination of catechin and immune protein provides a theoretical guarantee for research and development of health foods and the development of medical immunity.

## Figures and Tables

**Figure 1 foods-12-00462-f001:**
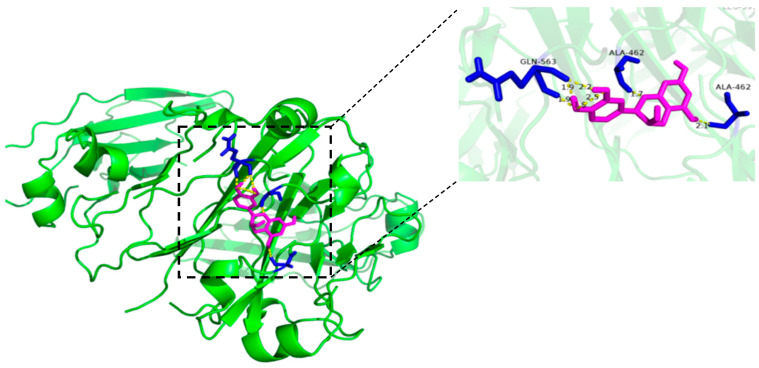
Molecular docking diagram of the catechin and IgY interaction.

**Figure 2 foods-12-00462-f002:**
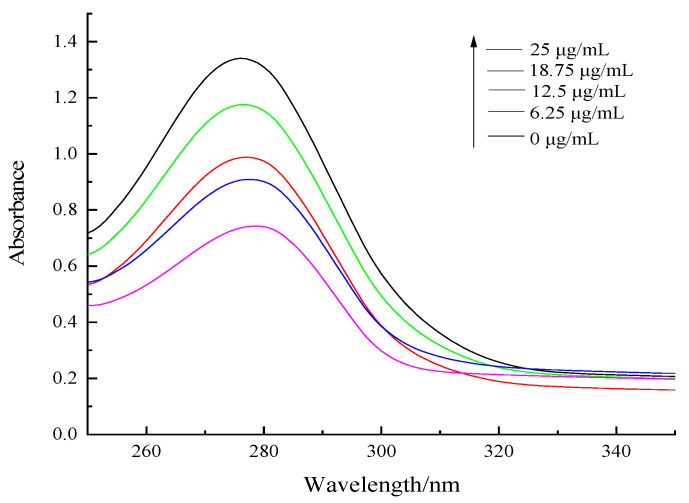
UV–vis spectrum of different concentrations of catechins binding to IgY.

**Figure 3 foods-12-00462-f003:**
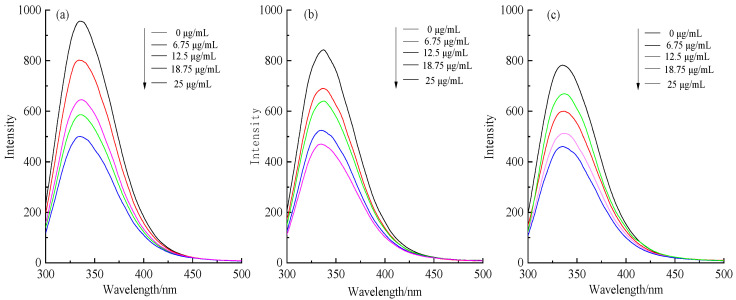
Fluorescence spectra of different concentrations of catechins binding to IgY. (**a**) 18 °C, (**b**) 25 °C, (**c**) 37 °C.

**Figure 4 foods-12-00462-f004:**
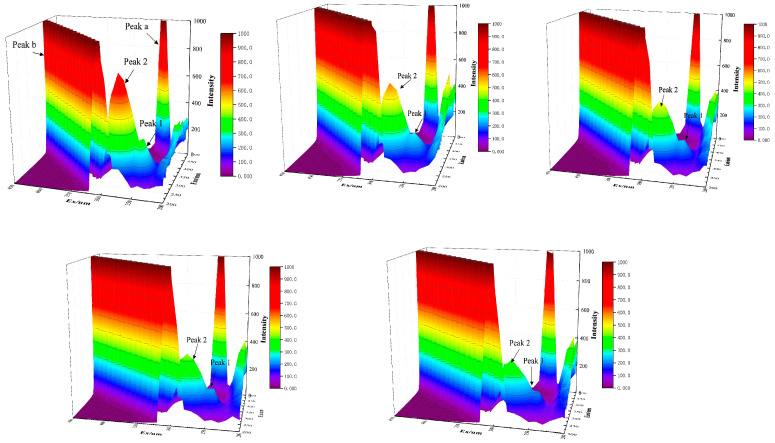
Three-dimensional fluorescence spectra of the interaction between catechins and IgY at different mass concentrations.

**Figure 5 foods-12-00462-f005:**
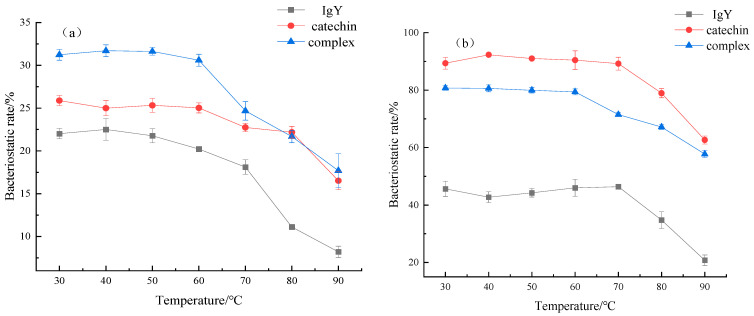
The effect of IgY interactions with catechins on antibacterial thermal stability of *E. coli* (**a**) and *S. aureus* (**b**).

**Figure 6 foods-12-00462-f006:**
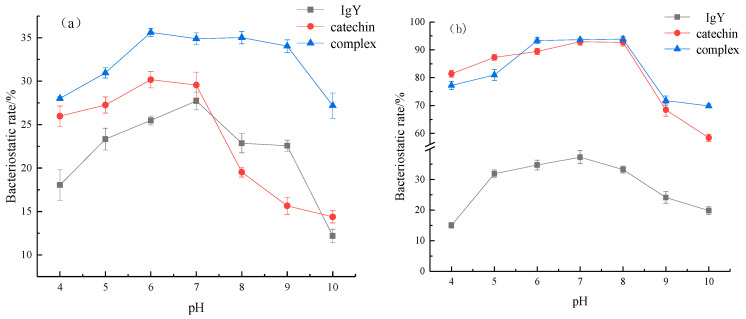
Effect of IgY interacts with catechins on antibacterial acid–base stability of *E. coli* (**a**) and *S. aureus* (**b**).

**Figure 7 foods-12-00462-f007:**
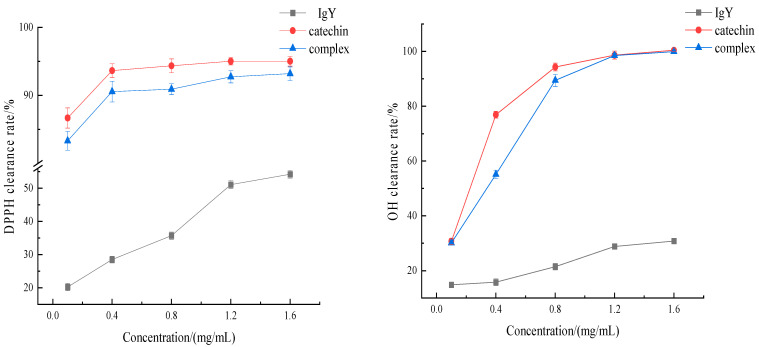
Effect of IgY interactions with catechins on antioxidation properties.

**Figure 8 foods-12-00462-f008:**
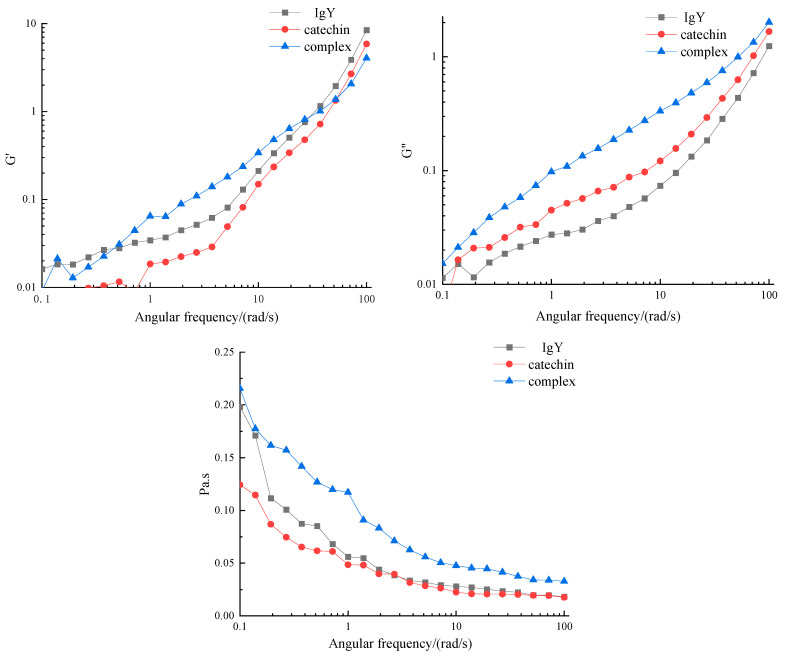
Effect of IgY interactions with catechins on rheological properties.

**Figure 9 foods-12-00462-f009:**
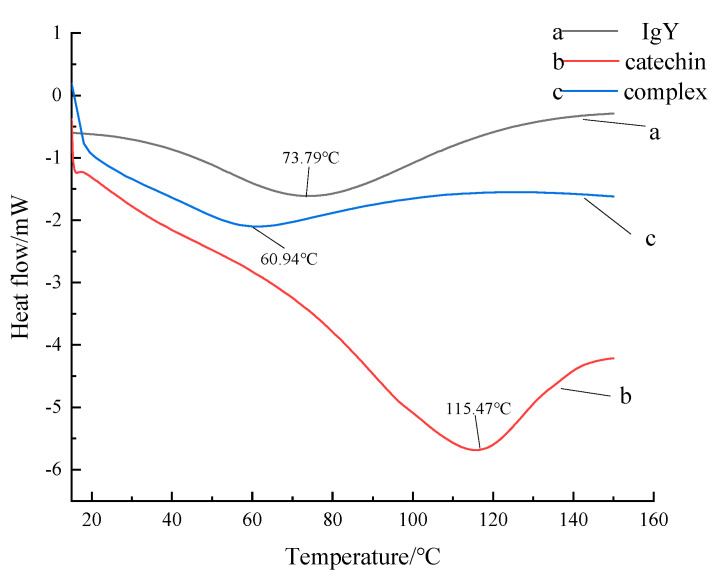
Differential scanning calorimetry analysis.

## Data Availability

The data presented in this study are available on request from the corresponding author.

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
