# Peer review of "Effect of Catechin on Yolk Immunoglobulin Structure and Properties: A Polyphenol–Protein Interaction Approach"

_foods, 2023, doi:10.3390/foods12030462_

Round 1
Reviewer 1 Report
I reviewed the manuscript entitled, Effect of catechin on yolk immunoglobulin structure and properties: A polyphenol-protein interaction approach. The study is well planned and executed in a systematic way. However, authors should consider below suggestions.
Abstract
Background of the study should be introduced before introducing research objectives
Research objectives are not clear. Please revise to identify easily
Keywords should be revised. All the keywords are matching with the words in title
Introduction
Citation format is not according to the journal format
Line 62: Whey Protein Isolation? whey protein isolate?
Line 64: Tea polyphenols should be revised as tea polyphenols
Line 68: scientific name must be in Italics
Line 80: Three-dimensional should be revised as three-dimensional. Authors should avoid using unnecessary capital letters in the middle of text unless it is necessary.
Line 94: Salicylic acid..s should be lowercase letter. Likewise, please revise throughout the manuscript
Line 96; Escherichia coli and staphylococcus aureus were from the microbiology lab teacher. Please revise it. Microbiology lab teacher? This is not the correct way of academic writing
Section 2.4. Please write the name of the UV instrument, model, company name, location, city and country. Also, provide the reference
Section 2.5. please write the name of Fluorescence spectroscopy instrument, model, company name, location, city and country. Also, provide the reference
Section 2.7.2. OH free radical scavenging rate: Please the citation
Section 2.11. Rheological analysis: please write the name of the rheometer used, model, company name, location, city and country. Also, provide the reference
Include the section statistical analysis
Line 182: The result obtained was shown in Figure 1.. can be revised as The results are shown in Figure 1
Section 3.1. and 3.2. There is no discussion and comparison with available literature. This section is very weak
Figure 2. legend lines look similar. I suggest to revise and change legends to different color or symbols
Figure 3. legend lines look similar. I suggest to revise and change legends to different color or symbols
3.3. Fluorescence spectroscopy: discussion should be improved and compare with available literature
Figure 4. axis of Figure 4 (X, Y, Z) is unreadable and I don’t understand it because of low clarity. Please improve the quality of the Figure 4
Line 284: escherichia coli, E should be capital letter and staphylococcus, S should be capital. Please revise the same in Figure 6
Section 3.7. please improve the discussion and compare with literature
Section 3.9. please improve the discussion and compare with literature
References are not according to the journal format. Please revise it point-by-point
Author Response
Dear Editors and Reviewer:
Thank you for the reviewers’ comments concerning our manuscript entitled “Effect of catechin on yolk immunoglobulin structure and properties: A polyphenol-protein interaction approach” (foods-2108724). Those comments are all valuable and very helpful for revising and improving our paper. We have studied comments carefully and have made correction which we hope meet with approval. The main corrections in the paper and the responds to the reviewers’ comments are as following:
Please see the attachment.

Reviewer 2 Report
The manuscript mainly focused on the effect of phenolics on functional properties of IgY, and the molecular docking and 3D spectroscopy were performed for better evaluating the properties of complex. The manuscript is well planned, designed and written. However, some minor modifications are needed.
1- Line 99-106: Please check the style. It is different from the whole text.
2- Line 123: Please change the unit of temperature. It was stated as celcius in some places. Use the same unit in the text.
3- Line 134: Please check the sentence.
4- Line 134-143: The methodology should be revised. There are grammatically errors and no explanation was found for the LB and OD.
5- Line 146: The sentence is not appropriate for the begining of the method section. The same situation is also observed for the Line 151. Please check the whole text and correct.
6- Line 152; Line 157; Line 158: Please give a space between the numbers and units. In some places, authors wrote this situation as true, but sometimes they wrote it wrong. The true version is 1 mL, 0.5 mg/mL.
7- Line 159-166: Please give more detail about this analyses. How did authors prepared the related solutions of FeSO4, salicylic acid etc.?
8- Line 172-176: What was the measuring geometry, cone-plate or plate-plate? Please mention the temperature at which the anlaysis performed. In addition, please describe the linear viscoelastic range of the samples. Did authors perform the tests at linear or non-linear viscoelastic range?
9- What about the statistical analyses? Please add a section and describe in detail the performed statistics.
10- Line 226: Please specify the temperature as celcius.
Author Response

(The authors gave the same response as above.)

Round 2
Reviewer 1 Report
The quality of the manuscript is now improved after adding the reviewer suggestions. In my opinion, this version can be accepted for publications.